# Design of a Pediatric Rectal Ultrasound Probe Intended for Ultra-High Frequency Ultrasound Diagnostics

**DOI:** 10.3390/diagnostics13101667

**Published:** 2023-05-09

**Authors:** Maria Evertsson, Christina Graneli, Alvina Vernersson, Olivia Wiaczek, Kristine Hagelsteen, Tobias Erlöv, Magnus Cinthio, Pernilla Stenström

**Affiliations:** 1Department of Clinical Sciences, Lund University, 22185 Lund, Sweden; 2Department of Biomedical Engineering, The Faculty of Engineering, Lund University, 22185 Lund, Sweden; 3Department of Pediatrics, Clinical Sciences, Lund University, 22185 Lund, Sweden; 4Department of Pediatric Surgery, Skåne University Hospital Lund, 22185 Lund, Sweden

**Keywords:** anorectal conditions, diagnosis, Hirschsprung’s disease, pediatrics, probe, ultra-high frequency ultrasound

## Abstract

It has been shown that ultra-high frequency (UHF) ultrasound applied to the external bowel wall can delineate the histo-anatomic layers in detail and distinguish normal bowel from aganglionosis. This would potentially reduce or lessen the need for biopsies that are currently mandatory for the diagnosis of Hirschsprung’s disease. However, to our knowledge, no suitable rectal probes for such a use are on the market. The aim was to define the specifications of an UHF transrectal ultrasound probe (50 MHz center frequency) suitable for use in infants. Probe requirements according to patient anatomy, clinicians’ requests, and biomedical engineering UHF prerequisites were collected within an expert group. Suitable probes on the market and in clinical use were reviewed. The requirements were transferred into the sketching of potential UHF ultrasound transrectal probes followed by their 3D prototype printing. Two prototypes were created and tested by five pediatric surgeons. The larger and straight 8 mm head and shaft probe was preferred as it facilitated stability, ease of anal insertion, and possible UHF technique including 128 piezoelectric elements in a linear array. We hereby present the procedure and considerations behind the development of a proposed new UHF transrectal pediatric probe. Such a device can open new possibilities for the diagnostics of pediatric anorectal conditions.

## 1. Introduction

Hirschsprung’s disease (HD) is a congenital disease with an incidence of 1:5000 in newborns [1]. It is characterized by a lack of ganglion cells in the bowel wall (aganglionosis), requiring surgical removal of the affected intestine [2]. Ganglia cells are normally found in the bowel wall’s submucosa and myenteric layer [3]. For the diagnostics of HD, a rectal biopsy is taken from the rectum, but there are reports of physical and psychological biopsy-related problems, as well as insufficient accuracy, in up to 50% of cases [4,5,6,7]. About 10 times more children than those just with HD, e.g., children with bowel dysmotility, need to undergo investigation by rectal biopsy [7,8,9]. Finding an instantaneous and secure diagnostic technique for HD to replace rectal biopsy could benefit this patient group enormously.

For replacing biopsy diagnostics, ultra-high frequency (UHF) ultrasound is undergoing research. Histo-anatomic morphometrics have been shown to differentiate between the aganglionic and ganglionic bowel wall [10]. Furthermore, differences have been shown to be replicated on UHF ultrasound (center frequency 50 MHz) imaging of bowel specimens [4]. These studies were performed with the aim of differentiating between aganglionic and ganglionic bowel in children who had already been diagnosed with HD. For primary diagnostics, with the aim of replacing the need for rectal biopsy, transanal and mucosal ultrasound imaging is required. The only UHF ultrasound probes currently available for commercial use today (UHF 48 with bandwidth 20–46 MHz and UHF 70 with bandwidth 29–71 MHz, FUJIFILM VisualSonics, Toronto, Canada) are too large for a child’s anus and rectum. A pediatric rectal UHF probe would enable the examination of large cohorts of children. Furthermore, it would facilitate the validation of UHF ultrasound by collecting images of reference bowel wall from healthy children. For other clinical conditions, a pediatric transrectal UHF probe could also facilitate detailed diagnostics of anorectal fistulas, internal and external sphincter injuries and pelvic floor malformations, and aid in the delineation of anorectal tumors. 

Our overall aim was to enable diagnostics with UHF ultrasound, but also to open new possibilities for the high-resolution ultrasound imaging of other anorectal and pelvic floor conditions. The specific goal of this study was to establish a requirement specification and suggest prototypes of a rectal UHF ultrasound probe suitable for use in infants. We hereby present the procedure and considerations behind the development of a proposed new pediatric UHF transrectal probe.

## 2. Materials and Methods

### 2.1. Settings

The study was performed at a department of pediatric surgery, which was appointed in 2018 as a national center for HD and anorectal malformations. It covers a geographical area of 5 million residents. The process underlying the development of the pediatric UHF probe followed a specific structure and involved an expert study group comprising pediatric surgeons (*n* = 5) and researchers within biomedical engineering (*n* = 4). The pediatric surgeons had 8–13 years of specialist experience and were subspecialized within the field of pediatric gastrointestinal surgery. The biomedical engineers and researchers were all specialized in medical ultrasound imaging and had considerable experience in working with UHF ultrasound. One of them also had experience in the life science development of rectal ultrasound probes for adults.

The development procedure was structured according to the following steps:Identification of probe requirements, according to anatomic, clinical and technical considerations;Review of available and, for the purpose, feasible, probes currently on the market and in clinical use;Sketching potential UHF ultrasound transrectal probes;3D prototype printing;Evaluation of the prototypes.

#### 2.1.1. Anatomic, Clinical and Technological Considerations

Anatomic considerations: The proposed probe was targeted for use in children weighing 3–15 kg. Since rectal ultrasound examinations were intended to be performed without the use of anesthesia or sedatives, patient comfort and safety were key considerations. Therefore, the metrics of pediatric anatomic anorectal anatomy, with regard to anal opening size, rectal diameter and rectum length, were collected from the literature. They were also collected from the medical charts of the last 10 patients treated at the department of pediatric surgery who had undergone anography diagnostics for HD [9]. Close contact between the probe head and the bowel wall to avoid shadowing effects within images was taken into consideration with regard to the proposed probe’s anatomic design.

Clinical considerations: Information about clinicians’ requirements for a transrectal probe was collected within the expert group and from discussions and clinical observations. Various aspects of patient safety, with regard to movement, environmental circumstances during anal examinations, and the examiner’s ergonomics, were taken into consideration when developing the design.

Technological considerations: Technical specifications, based on previous knowledge of UHF ultrasound of the bowel wall, were collected within the expert group. A small-sized pediatric anal probe still needed to include technical prerequisites for encompassing UHF ultrasound technology. The probe center frequency, the number of piezoelectric elements, and the image view were all taken into account. The contact between the probe head and the bowel wall was also contemplated with regard to technical solutions. Electronics requirements were noted for a later evaluation.

#### 2.1.2. Review of Available and Feasible Probes

An analysis of the market was performed to identify currently available probes suitable for use regarding our aim to introduce UHF technology in the form of a pediatric anorectal probe. The overview was intended to cover various probes’ physical dimensions, frequency ranges, number of elements and the field of view. Clinical site visits to departments using some small-sized ultrasound alternatives in diagnostics were made in order to provide detailed clinical insights into their use in practice.

#### 2.1.3. Sketching 

The identified parameters of the anal tract of importance for the probe’s design and size were the diameter of the anal opening, inner diameter of the rectum, and insertion length. All measurements were set to a minimum in order to avoid discomfort for the child, while still enabling the allowance of the identified technical prerequisites. Based on these results, probe prototypes were sketched using a CAD (computer-aided design) program, PTC’s Creo Parametric 7.0.2.

#### 2.1.4. 3D Printing of Probe Shell Prototypes

Prototype probe shells were printed using a Formlabs 2 (Formlabs, Somerville, MA, USA) printer and stereolithography. For biocompatibility, BioMed White (Formlabs, USA) was used as a resin polymer. A thickness layer of 50 µm was used, resulting in a total of 3414 layers for each probe prototype. In post-printing processing, the 3D prototypes were placed in isopropanol (75%) for 5 min, cured for 60 min in an ultraviolet oven (Form Cure [Formlabs, USA]) at 60 °C. They were then placed again in a 75% isopropanol bath for 2 min followed by a new cure round at 60 °C in an ultraviolet oven for 20 min. In a final post-processing step, any residuals from the resisting support material were scraped away, and the entire probe was sandpapered.

#### 2.1.5. 3D Prototype Testing

The feasibility of each of the developed 3D-printed prototype probe shells was tested by five pediatric surgeons of both genders. Their preferred grip of the prototype, as well as their opinion on the design and overall handling, were evaluated according to a scheme covering stability, hand movement flexibility and ergonomics in the preferred examination positions.

## 3. Results

### 3.1. Anatomic, Clinical and Technological Considerations 

Anatomic considerations: Anal opening size: Sizes of anal openings in newborns up to 1-year-olds, with or without malformations or aganglionosis, were reported in the literature to be 12–15 mm [11]. According to the medical charts held by the department of pediatric surgery, anal openings of children with HD undergoing washouts (*n* = 10) or children with anorectal malformations with perineal fistulas (*n* = 10) undergoing calibrations and enemas were 6–10 mm. The diameter of the rectal suction instrument (rbi2^®^) used for diagnosing HD in children weighing 2.5–7.1 kg was 7 mm, which entered the anus and rectum without any resistance.

Rectal diameter: The rectum width was assessed in anographies in children weighing a median of 4.8 kg (range, 3.1–7.1 kg) undergoing examination for HD (*n* = 10) at the department of pediatric surgery. Five children had rectosigmoid aganglionosis and five had no aganglionosis according to the pathology biopsy report. Their median inner rectal diameter was 14 mm (range 8–25 mm). In the children who underwent pre-operative rectal wash-out prior to surgery for HD, a Foley catheter with a corresponding median size of 7–11 mm was used. 

Length of rectum: In order to plan the insertion depth of the proposed probe, the length of the rectum, as measured from the skin as well as 1 cm above the dentate line up to the start of the sigmoid curve, was assessed in the five infants undergoing wash-outs prior to surgery for HD. The length of the rectum, assessed by inserting a soft Foley^®^ catheter, was a median of 50 mm (range, 40–65 mm) before resistance occurred. The length of the rectal suction instrument (rbi2^®^) in children weighing 2.5–7.1 kg reached up to maximum of 6 cm from the skin verge. A schematic sketch of the anorectal dimensions defined in this study is shown in Figure 1.

Clinical considerations: With respect to the design of the proposed probe, clinicians’ requirements for patient comfort, safety and ergonomics were focused on the size and shape of the probe’s head, shaft, and handle. The probe head should be small enough to enter the anus without causing discomfort to the patient. The shaft’s length and size should allow flexibility of the probe element direction so that the probe head can reach the mucosa posteriorly, anteriorly and on the lateral sides. The handle should enable the examiner’s hand to rest on the underlay in order to eliminate movement when the image is taken. Furthermore, it should fit most surgeons’ hands, regardless of their size. The handle and shaft should enable the examination to proceed easily while the examiner stands right in front of the child, as well as on the side of the child. The size and grip of the handle were tested by the surgeons, resulting in a requirement for a handle length of 9–11 cm and a circumference of 4–7 cm (Sketch in Figure 2). 

Technological considerations: An ultrasound resolution (frequency) sufficient to replicate all layers of the rectum wall, including the mucosa, submucosa, muscularis interna, muscularis externa and the serosa, was requested.

Center frequencies: Since the rectal wall in children has been shown to be approximately 0.8–2 mm thick and histo-anatomic bowel wall layers extend to 0.2 mm [4,10], the consensus was that visualization of such small anatomy required UHF ultrasound of 30–50 MHz center frequency. 

Linear array probe or rotating single-element probe: The probe type used in previous UHF ultrasound studies of examinations of bowel wall ex vivo and in vivo from the serosa was a linear array probe (UHF70, FUJIFILM VisualSonics, Toronto, ON, Canada). A linear array probe produces rectangular images, i.e., field of view, and is often designed to operate at high center frequencies and image superficial structures. It also produces images with both good axial and lateral resolutions. In contrast to this, a rotating single-element probe, as is used in intravascular UHF ultrasound imaging with a circular rotating piezoelectric element probe, has the advantage of creating 360° circumferential images. The downside with a single probe element is diverging ultrasound beams, resulting in poorer lateral resolution compared to a linear array probe, especially at larger depths/distances. Based on these facts, it was decided that a linear array probe was preferable, especially since a higher lateral resolution was favored compared to circumferential imaging. 

Longitudinal or transversal imaging: In previous bowel wall examinations using UHF ultrasound, only longitudinal ultrasound imaging with a 256-element linear array probe has been studied (UHF70, FUJIFILM VisualSonics, Toronto, ON, Canada) [4,10]. As a result of this experience, longitudinal imaging was the first choice by the expert group. However, transversal imaging is the most commonly used imaging approach in the clinic. Transversal imaging may be more intuitive to interpret and understand, easier to orient in and, if probe elements are arranged along a curved probe head, a so-called curvilinear array, then it is more beneficial for investigating deeper tissues since a widening image area with depth is obtained. Another approach discussed, enabling both longitudinal and transversal imaging approaches, was a moving element linear array probe delivering a longitudinal image view, collecting 2D images, and turning them into 3D imaging volumes. However, this technique would be sensitive to motion artifacts, which are likely to occur in a baby who is awake during the examination. Since the time span for the elements being in motion is short (approximately 1–3 s for a 90° angle), the possibility of using such a probe was not excluded. 

Number of probe elements: More elements deliver a larger field of view but require a larger area on the probe. However, in previous research, where 256 elements were used for diagnostic UHF ultrasound imaging of the bowel wall, only less than half of each available image was required for accurate measurements. Therefore 128 elements were considered to suffice. A smaller number of probe elements, delivering a smaller area, would also increase the likelihood of obtaining good contact between the probe and the mucosa. This, in turn, would reduce the risk of shadowing effects as a result of the presence of air between the probe and the bowel wall. Still, a drawback of a single-element probe is that it must be driven by special motors, which require space and other technology.

Probe contact to bowel wall: To avoid shadowing, air between the probe head and the bowel wall should be limited. Close contact between the probe head and the bowel wall could be obtained by using a larger head in order to fill the rectum’s lumen, by the use of a bent, or bendable, probe head. Alternatively, a water-filled balloon surrounding the head could be used, pressing it towards the bowel wall (see endobronchial ultrasound probe, Table 1). It was decided that a straight probe head, without a surrounding balloon, would be both desirable and possible if the stiff shaft could lean against the mucosa in line with the suction biopsy instrument. A smaller area of a linear array probe, including a few elements in combination with an easily maneuverable design, could still enable close contact with the mucosa.

### 3.2. Review of Available and Feasible Probes 

A review of various types of ultrasonography using small probe heads, operating at high center frequencies, was explored in the literature and in manufacturers’ product information. This information is collated and presented in Table 1. According to the review, the majority of ultrasound probes in clinical use today are of too large a size for rectal imaging in infants. Alternatively, they deliver too low center frequencies (only around 5–18 MHz) compared to those desired for a pediatric UHF ultrasound probe. Additionally, some of the ultrasound probes identified and observed had very long shafts intended for examinations of tissues at greater distances (such as the heart through the esophagus or the bronchus through the trachea) than the bowel wall anorectally in children. Although the sizes and designs of the heads were of interest, longer probe shafts are not desirable for use in children if they are awake during the examination, because the closer the hand can be to the targeted tissue, the less likelihood there will be of artefacts developing if the patient moves. 

Three types of ultrasound probes with specific prerequisites and suitable pediatric dimensions were selected for observation in clinical use. For imaging the lumen of small blood vessels, the high-frequency intravascular ultrasound (IVUS) probe was observed. The IVUS probe is a catheter-based type small enough for use in children, and with a center frequency of 50 MHz, which most likely would be sufficient to view the histo-anatomic layers of the bowel wall. The downside to using a catheter-based ultrasound device in the rectum is that it does not reach to the mucosa circumferentially. Therefore, a large amount of gel will be needed to fil the rectum, in order to avoid air accumulating between the probe and the mucosa. 

For imaging cardiac details in small children, the transesophageal probe with linear array elements was inspected, as used prior to cardiac surgery. For imaging details of lung bronchials, the endobronchial ultrasound (EBUS) was observed during bronchoscopy. Both these latter two probes consist of a number of probe elements next to each other (similar to the UHF ultrasound probe used in our project on bowel wall so far) and were also considered to be suitable in size and design for pediatric anorectal use. The observation area was considered to be large enough, but their frequency ranges (3–8 MHz and 5–12 MHz, respectively) were too low for the specified requirements (Table 1).

### 3.3. Sketches of Probe Shell Prototypes

Suggestions and sketches of two probes were produced, based on the information on suitable size, form, and allowance of a technology including 128-linear elements (Figure 2). The probe heads developed were, according to the collected information, straight and set to sizes of 7 mm and 8 mm, respectively, to suit most infants. To make the probe more maneuverable, a narrowing of the shaft was sketched in one of the suggested prototypes (5 mm shaft diameter). The length between the head and the handle was set to be close to 10 cm. This was in order to allow both flexibility and stability, and for patient comfort and safety. To enable many types of grips and to make the probe easy to handle, the designs of the prototypes’ handles were inspired by the shape of a screwdriver but were somewhat larger. 

### 3.4. 3D Printing of Probe Shell Prototypes

Two 3D probes were printed according to the sketches and the methods described above. The proposed 3D probes prototypes are shown in Figure 3. 

### 3.5. 3D Prototype Testing

Five pediatric surgeons—three female and two male—tested the two probe prototypes on a baby model. The larger probe, which had a size of 8 mm of both the head and neck, was preferred independently by all surgeons. The reasons stated were that both the design and size facilitated stability and ease of insertion, compared to the somewhat thinner probe prototype with a 7 mm head and a 5 mm shaft (Figure 3). The 8 mm probe prototype allowed the surgeon to hold it comfortably using a one-hand grip. Multiple one-hand grips and working positions for the surgeons were evaluated. Some examiners chose to stand at the feet of the baby during the examination, whereas others preferred to stand by the baby’s side. Some preferred to hold the prototype probe very close to the baby, while others preferred to hold it 5–10 cm from the baby’s anus. The 8 mm model was also preferred with regard to ergonomics and positioning (Figure 4). 

A tactile indicator was printed on the front of the handle of the probe prototype to enable indication of the position of the probe elements. The surgeons’ feedback was that this was too small to recognize under the plastic protection wrapped around the probe. The need for enlargement and visibility in updated prototype versions was advocated in order to ensure the correct positioning of the probe.

## 4. Discussion

The aim of this study was to create a geometrically designed requirement specification for a pediatric UHF rectal ultrasound probe in order to enable high-frequency anorectal ultrasound examinations. With information gathered on anorectal measurements and for technology prerequisites, sketches and 3D-printed prototypes were produced and tested. The prototypes selected were reported to be satisfactory with regard to their design and ergonomics. However, some features were missing, such as markings on the handle or neck, to aid the examiner in orientating the probe. During the development process, specific technical uncertainties were raised, especially regarding the type of elements, and the highest center frequency for these, used within a small-diameter probe. The main challenge was to develop a design for a probe that was small and neat, but that still met all technical specification requirements. With this achieved, still bearing in mind that electronic prerequisites need to be considered, this is a first step in the development of a pediatric UHF rectal probe—a product that is urgently required for our pediatric population. 

### 4.1. Design

The pediatric anorectal probe was intended to be used primarily in newborns, since this is the time when the need for an instant diagnostic method for HD and detailed fistula exposure in anorectal malformations is the greatest. As a result of the neonatal consideration, it was decided that the maximum head size should be 7–8 mm. This size was expected to be appropriate, or at least not too large for the majority of patients, since most newborns have an anal opening of a maximum of 12 mm. A small-sized anorectal probe can also be useful in older children. Still, a larger probe head would probably facilitate the connection between the head and the rectal mucosa, which would limit air-induced shadowing effects in the ultrasound images, and the need for gel in the rectal space. Additionally, more advanced technology and electronics could be used if the probe head was larger. However, shadowing effects could be avoided if using only a small area of linear array elements. This is feasible in the examination of a smaller specific area, such as that of the bowel wall instead of aiming for a circular image. In addition to size consideration, and prioritizing comfort for children of low weights, it was decided that the smallest size of probe was preferable. In testing the prototypes, the larger probe with a size of 8 mm of both the head and shaft was preferred by all surgeons in our study, since this design and size facilitated stability and ease of insertion. The bigger geometry could also be more beneficial when accommodating the electronics within. The size of the rectal suction biopsy instrument (rbi2^®^) is 8 mm, which causes only limited tension at the anal opening for most children. Nevertheless, children with fistulas within anorectal malformations could have anal openings narrower than 8 mm before surgery. With regard to inserting electronics, it might be difficult to create a linear high-frequency probe for anal openings smaller than 8 mm. This remains to be clinically evaluated in the future.

A total probe shaft length was tested by the grip of other instruments and the optimum length was considered to be 10 cm. The length was decided as a trade-off between greater measuring distances provided by longer shafts, e.g., in older children with a longer rectum, and the increased stability given by shorter shafts. The 10 cm shaft length was appreciated by all the surgeons in our study, and no surgeon wished that the length was longer. This shaft length also allowed a stable examination of tissues at a shorter distance from the anus, when holding the probe by the neck, letting the handle rest on the upper side of the hand (Figure 4). The testing revealed satisfaction, especially with regard to grip and a flexible examination, and with being able to hold the probe in a position according to the surgeon’s personal preferences, i.e., either standing at the patient’s feet or by their side. In particular, the handle’s fit in smaller hands was appreciated, allowing easy handling by both genders. 

Regarding the angulation of the probe head, this implied the possibility of visualizing specific areas more closely without the need to insert gel into the rectum. However, angulation is not highly desirable, because if the shaft is flexible, then this probably places more demands on technology and electronics. If it is rigid (as in the hockey stick design described in Table 1), then this requires a larger anal opening diameter. During the clinical observation of the transesophageal probe, no angulation of the probe head was required to provide sufficient contact between the probe elements and the esophageal wall. Therefore, a straight probe head–neck was suggested. However, to ensure that a straight head-neck probe with no angulation is the best design for in vivo measurements in the rectum, clinical evaluations are needed. 

### 4.2. Technology 

Regarding the probe center frequency, 50 MHz was the level set in this study since this has been shown to be useful in delineating between histo-anatomic layers of the bowel wall. The 50 MHz frequency provides resolution of down to 30–50 µm [12]. However, such high frequencies do not allow imaging of deeper depths and, therefore, not all the layers of the bowel wall. Ultrasound frequencies used clinically today usually exhibit a much lower range (2–15 MHz), resulting in a resolution of a maximum 100 µm [13] and an imaging depth of more than 10 mm. In a pediatric cohort, lower frequencies of 15–20 MHz, corresponding to an imaging depth of approximately 2 cm, are already in use, as for visualizing pelvic floor components. These are, however, not suitable for the detailed diagnosis of pathologies of the bowel wall, e.g., aganglionosis, or for providing in-depth information about the internal sphincter. For such complex investigations, a center frequency of 30–50 MHz is desirable, resulting in an imaging depth of a maximum of 5–10 mm. 

To produce a probe with a large bandwidth, covering 25–70 MHz, and which allows imaging at both the lower and higher frequency spans, would be desirable, but it is not currently known if this would be possible. However, the DualproTM IVUS + NIRS probe (Table 1) delivers a frequency span of 30–65 MHz and has a fractional bandwidth of 60%, but its electronics and probe elements differ greatly from those of our ideal array probe. 

In conventional ultrasound examinations within the lumen, e.g., rectally and intravascularly, transversal ultrasound images (cross-sectional images) are used most commonly. In contrast to this, in our prior research on the use of UHF ultrasound for discriminating between the aganglionic and ganglionic bowel, a linear array probe producing longitudinal ultrasound images was used [4]. The benefit of longitudinal images is that a larger area of the bowel wall viewed caudally to cranially can be investigated within the same image. A cross-sectional view could, of course, be an option, but investigation as to whether differentiation between ganglionic and aganglionic bowel wall can also be made in cross-sectional images is needed and studies are currently ongoing. 

If a transversal imaging approach (cross-sectional images) is preferred, then a curvilinear probe including probe elements arranged along a curved surface would probably be favorable. This is because a curve will allow more elements to be placed in the probe, which would be necessary since a cross-sectional view requires the mounting of elements perpendicular to the shaft, instead of as in the longitudinal case, i.e., in parallel. With such a solution, a decrease in lateral resolution will then be seen, and especially with depth since the ultrasound beams will slightly diverge from each other. A decrease in lateral resolution would, on the other hand, also be the result of a rotating single-element probe. This is since the beamforming will not be able to be performed with a one-element probe because of a diversion of beams. In contrast, a beamforming can be obtained with an array probe. However, if a pediatric rectal probe with circular and cross-sectional UHF imaging is the aim, then a lower lateral resolution might need to be accepted, as long as the resolution of the central image is sufficient. Whether this is good enough to be used for HD diagnosis could be studied, e.g., using the IVUS catheter probe. 

A great benefit of using the IVUS technique is that these probes already operate at UHF, which is required for HD diagnostics. Obstacles foreseen with the rectal use of IVUS are the diameter difference between the rectum and probe, and the fact that a stiffer shaft would be desirable to enhance the steering of the probe. 

Another option is to rotate a linear array probe within a rigid tube, as in the Endocavity 3D 9038, BK Medical probe (see Table 1). This probe would provide 3D images with high resolution, but the electronics required would most probably not fit into a small-sized pediatric ultrasound probe. 

The strengths of this development study are that it followed a participatory design [14] within a high-competence medical center performing unique translational studies of UHF ultrasound of the bowel wall. Additionally, anorectal measurements from radiology reports and clinical observations were collected within a national center with considerable experience in registering pediatric anorectal dimensions. Furthermore, the design specifications presented here are based primarily on patient safety and comfort. 

One certain limitation of our study was that all available probes on the market could not be identified. Therefore, some might not have been included in our review, or some that have just been launched onto the market could have been omitted. Another limitation was that the prototype has not yet been tested on a child. The reason for this was that this study was required to be performed in order to be able to obtain ethical approval for testing probes and prototypes anorectally in humans. A serious consideration is that the 8 mm probe head could be too small for manufacturers to choose to invest in financially, as a result of the difficulties and expenses in developing small electronic devices. 

### 4.3. Future Improvements and Work

This study mainly focused on the design aspects and on the technical specifications of a potential pediatric rectal probe. With these considerations now in place, the electronics required for creating an UHF ultrasound probe can begin. One clear improvement of the probe design will be to mark the shaft with depth distances and orientation markings indicating the head’s direction up and down. This would facilitate descriptions of examinations and imaging orientation. These indications would preferably be both visual, for example, using colored dots indicating the distance to the tip of the probe, as well as using distinct tactile markings large enough to enable detection through a sterile plastic wrap. 

Since existing ultrasound probes have similar designs to our prototypes, such as the Philips S8-3t transesophageal probe, and the endobronchial ultrasound BF-UC190F probe, we believe that the creation of a pediatric UHF rectal probe is feasible. However, we are aware that in order to insert the desired UHF technology into the probe, the demands on its construction and electronics will be high. The results of our study should be widely contemplated with respect to how they can be interpreted from the perspective of previous studies and of the working hypotheses. The findings and their implications should be discussed in the broadest context possible. Future research directions may also be highlighted.

## 5. Conclusions

This is the first step in developing a novel non-invasive diagnostic method to examine the bowel wall in children—a pediatric anorectal UHF ultrasound probe. We believe that such a device can open new possibilities for diagnosing anorectal conditions in children.

**Table 1 diagnostics-13-01667-t001:** Overview of selected probes available on the market and in clinical use for diagnostics.

Probe	Appearance	Applications	Frequency Range (MHz)	Probe Type/Field of View	Number of Elements	Physical Dimensions (mm)	Comments: General and Specifically Regarding Anorectal Use in Children
Transesophageal (TTE) S8-3t, Philips, Amsterdam, The Netherlands [15]	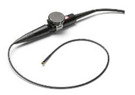	Cardiac imaging from the esophagus. Adapted for use in children.	3–8	Phased array/90° sector.	32	Head: 7.5 × 5.5 × 18.5. Shaft diameter: 5.2. Shaft length: 880.	+ Small-sized probe head + Enables angulation of probe head − Shaft too long to use anorectally in an awake child − Large handle—two hands are needed to rotate the dials on the handle that steer the probe − Too low frequency
Endobronchial ultrasound, EBUS, BF-UC190F, Olympus Medical Systems, Long Thanh, Vietnam [16]	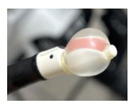	Pulmonary bronchus imaging.	5–12	Curved linear array/65° sector.		Head diameter: 6.6. Shaft diameter: 6.3. Shaft length: 600.	+ Small-sized probe head + Enables angulation of probe head + Enables taking biopsy + Water-filled balloon around probe head implying better tissue contact in air-filled environment − Shaft too long to use anorectally in an awake child − Too low frequency
Intravascular ultrasound (IVUS), Dualpro™ IVUS + NIRS, Infraredx Inc., Bedford, MA, USA [17]	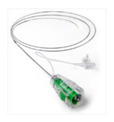	Imaging from the inside of the lumen of the vessels.	35–65	Rotating single element/ 360° image.	Single element	Head diameter: 0.8–1.15. Shaft diameter: 1.2. Shaft length: 1600.	+ 360° 3D images + High frequency—high resolution + Extended bandwidth, can transmit both lower and higher frequencies + Near infrared spectroscopy is integrated − Mismatch between rectum and probe diameter, risking insufficient contact and shadowing
Hockey stick L8-18I-RS, GE, Chicago, IL, USA [18]	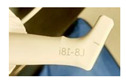	Shallow imaging, e.g., cardiovascular, musculoskeletal and small parts.	8–18	Linear array/ rectangular.			+/− Smaller size than conventional ultrasound probes, but too large to be used anorectally in children − Too short neck − Too low frequency
Laparoscopic L44LA, FujiFilm [19]	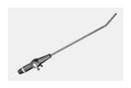	Imaging during laparoscopic surgery.	2–13	Linear array/rectangular, maximum width: 36 mm.		Head diameter: 10.	+ Small size of probe head + Angulation of probe head − Shaft too long to use anorectally in an awake child − Too low frequency
Endocavity, 3D 9038, BK Medical, Burlington, MA, USA [20,21]	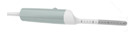	Transrectal and transvaginal imaging.	4–14	Rotating linear array/360° image.	192	Diameter: 16. Neck length: 155.	+ 360° image created with linear probe + 3D images can be produced − Too large to be used anorectally in children − Too low frequency
Prostate Triplane Transducer, 9018, BK Medical [22,23]	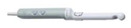	Transrectal probe for prostate imaging.	4–14	Curved linear array/180° sector.	128 + 192	Diameter: 20.	+ Imaging in two perpendicular planes + Enables taking biopsy − Too large to be used in anorectally in children − Too low frequency

## Figures and Tables

**Figure 1 diagnostics-13-01667-f001:**
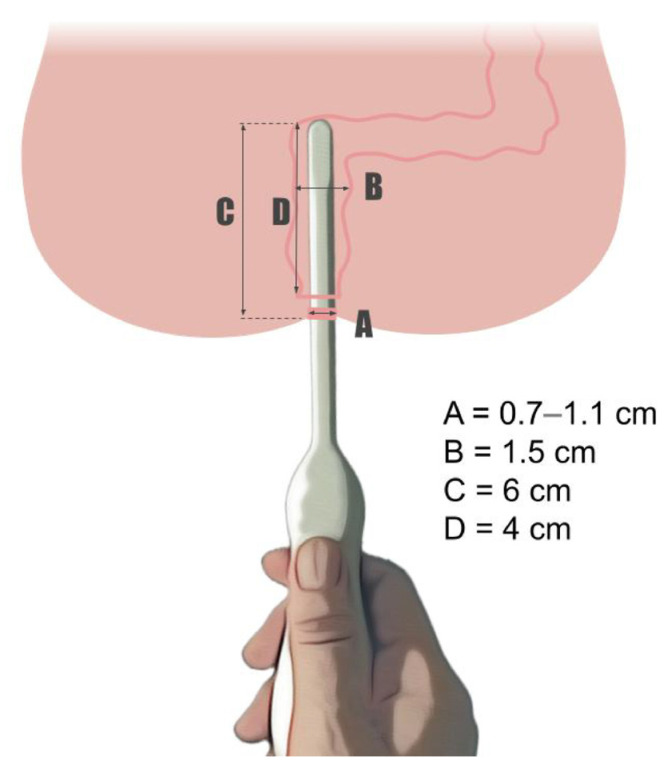
Illustration of a pediatric anorectal probe including both dimensional prerequisites according to clinical measures and the literature, as well as technical feasibility for ultra-high frequency technology. A. Feasible diameter of the probe inserted in the anus without causing any discomfort. B. Size of the rectum’s inner diameter. C. Length of the rectum from the skin to the sigmoideum. D. Distance from 1 cm above the linea dentata up to the sigmoideum.

**Figure 2 diagnostics-13-01667-f002:**
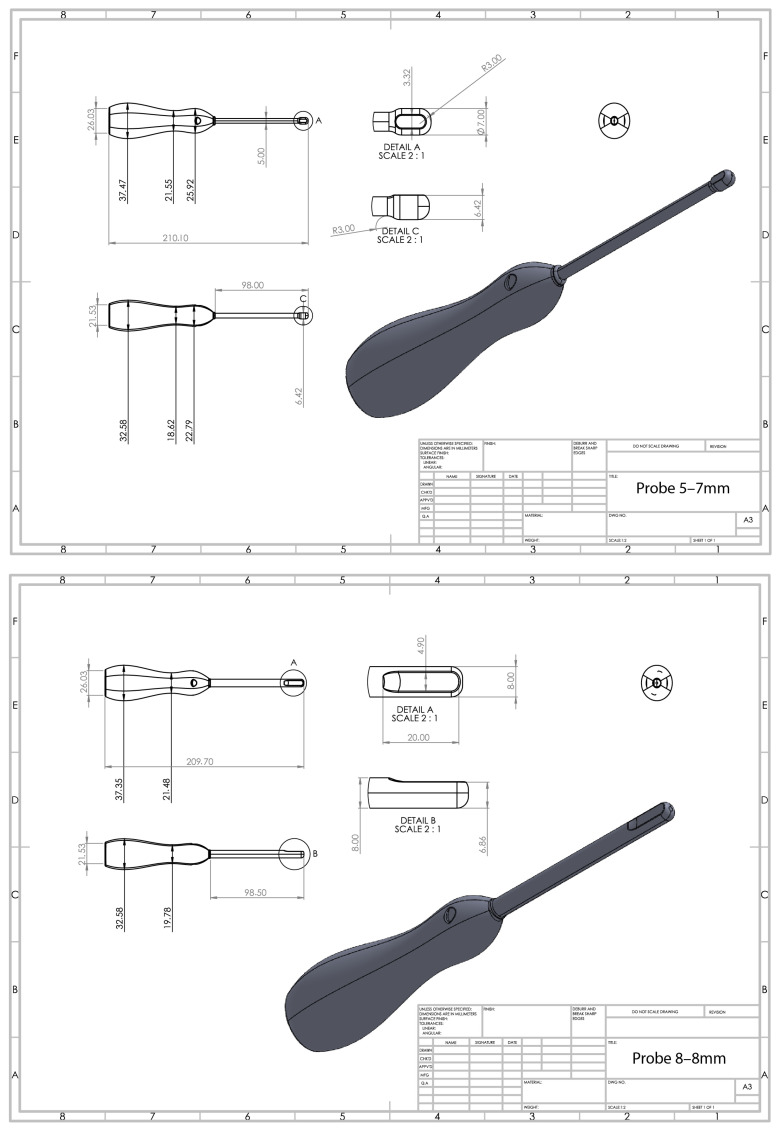
Sketches of two probe prototypes: Probe 5–7 mm with a thinner neck than head and Probe 8–8 mm, i.e., the same diameter for the probe’s head and neck Detail A shows where the probe elements would be positioned.

**Figure 3 diagnostics-13-01667-f003:**
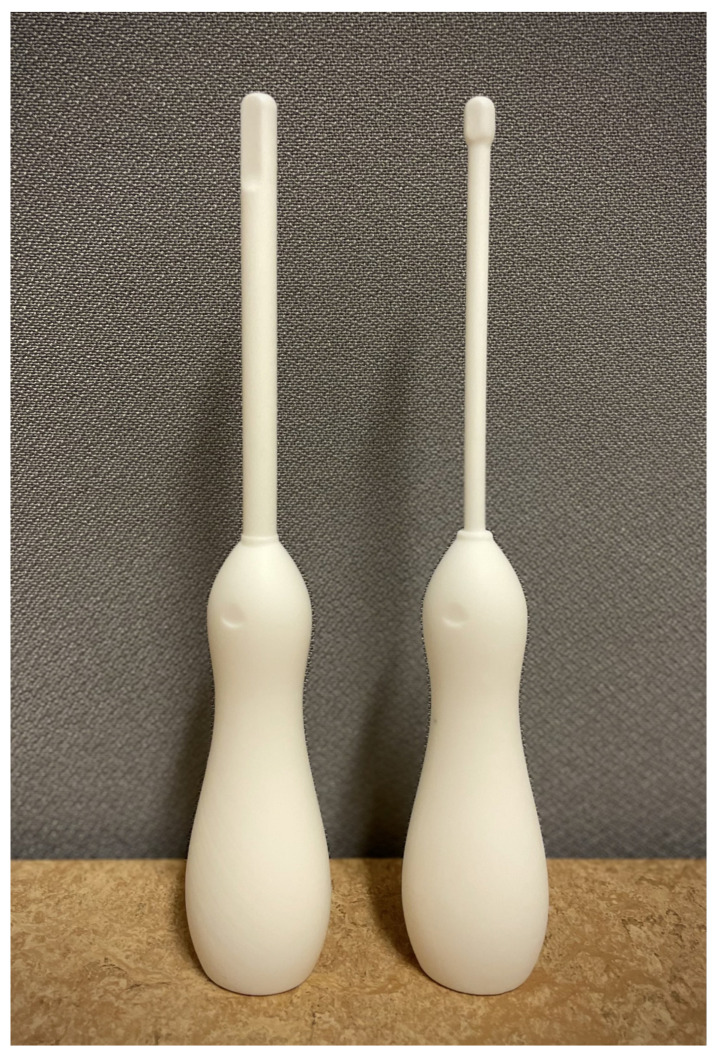
Two prototypes of a pediatric anorectal ultra-high frequency ultrasound probe. The prototype on the left has a head size of 8 mm and a shaft size of 8 mm. The prototype to the right has a head size of 7 mm and a shaft size of 5 mm.

**Figure 4 diagnostics-13-01667-f004:**
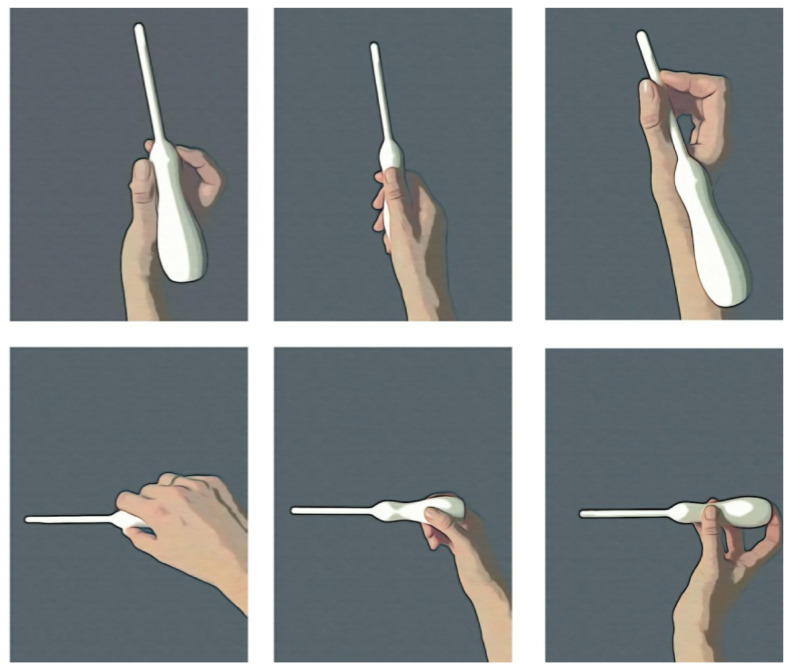
Overview of different grips used by pediatric surgeons of the ultra-high frequency ultrasound probe prototype. The grips shown in the top images were used by surgeons standing at the baby’s feet. The grips shown at the bottom were used by surgeons when standing at the side of the baby.

## Data Availability

The data presented in this study are available on reasonable request from the corresponding author.

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
