# Peer review of "Design of a Pediatric Rectal Ultrasound Probe Intended for Ultra-High Frequency Ultrasound Diagnostics"

_diagnostics, 2023, doi:10.3390/diagnostics13101667_

Round 1

Reviewer 1 Report

A well conducted study on the development of a pediatric endoanal high frequency ultrasound probe.  The paper might be shortened a bit e.g. information on the catchment population of the clinic seem superfluous since the probe must be intended for general use.   The numeration of titles and paragraphs could perhaps be omitted or simplified. As it stands now one expect that the paragraph number in methods will correspond to those in the results, which they do not.

I think another review of the manuscript by a native English person/doctor would improve readability and also have the potential to shorten the text a bit.

The beginning of the abstract is a bit  "heavy" . Why not write something like: It has been shown that ultra-high frequency ultrasound applied to the external bowel wall can delineate the layers in great detail and distinguish normal bowel from aganglionosis in Hirschprungs disease (HD).  Thus endoanal UHF might  be a very valuable tool for screening/diagnosing  HD in the newborn. Potentially this would reduce or lessen the need for biopsies that today are mandatory for the diagnosis.  However to our knowledge no suitable probes for such use are on the market. The aim....

Another suggestion would be to perform some UHF investigations ex-vivo from the mucosal side of the bowel to ascertain that the results obtained are comparable to those demonstrated when applying the probe from the outside.   This need not to be in the text, just a suggestion to do this, if not already done,  before too much work is invested in the endo-anal probe

Author Response

Dear Reviewer 1

We are grateful for the review you presented. We changed accordingly and think that manuscript improved much.

Best regards

Pernilla Stenström

Reviewer 2 Report

The aim of this study was to define the specifications for a rectal UHF ultrasound probe suitable for use in infants. The study includes a review of existing probes that partially meet such specifications and the 3D printing of two prototype probe shells that were used in ergonomics tests. As highlighted by the authors, this is the first step toward the development of a working probe, which should include the transducer elements and the cables connecting to a suitable scanner.

The word “transducer” should be substituted by “probe” either in the title and in the paper, because probe issues rather than transducer elements issues are discussed here.

The abstract needs a strong revision, by also considering the following hints:

-          Please specify which frequencies you consider UHF (>30 MHz?)

-          Lines 12 14: please rephrase “According to research” and “within research” (unclear)

-          Line 16: which is “the current UHF transducer”? Do you mean “the commercially available UHF probes”?

-          Was “The aim was to identify a UHF” probe? or (maybe better) “to define the specifications of an UHF probe for…”?

Specific comments:

P.2 L.54

Which is “The only UHF ultrasound transducer currently available today”? No reference is given.

P.2 L.78

What does “ME” mean?

P.2 L.87-88

Please rephrase point 3

P.3 L.116-117

Please rephrase

P.3 L.125

Substitute “scan” with “analysis” and rephrase the entire sentence

P.3 L.133-134

I’d substitute “Identified measures” with “Parameters”

P.3 Sec.2.6

I’d clarify that you implemented prototypes of the probe shell

P.6 L.222-226

A drawback of single element probes is that they must be driven by special motors

P.6 L.244-245

In which probe “only less than half of each image is used for analysis”? You mention a 256-element probe but do not provide any reference

P.11 L.371

Please remove “but not the other way around”

Table I

Please make efforts to reduce the size of such a Table

In general, the text could be improved, possibly through the help of a native English speaker. 

Author Response

Dear Reviewer 2

We are grateful for your review. We changed accordingly, and think that manuscript improved much. The Table 1 is shortened and hopefully more comprehendible. 

Best regards

Pernilla Stenström
